
# Pathfinding in stochastic environments: learning *vs* planning

Alexey Skrynnik[1,2,3], Anton Andreychuk[2], Konstantin Yakovlev[2,3] and Aleksandr Panov[1,3]

[1] Cognitive Dynamic Systems, Moscow Institute of Physics and Technology, Moscow, Russia
[2] Artificial Intelligence Research Institute AIRI, Moscow, Russia
[3] Federal Research Center "Computer Science and Control" of the Russian Academy of Sciences, Moscow, Russia

## ABSTRACT

Among the main challenges associated with navigating a mobile robot in complex environments are partial observability and stochasticity. This work proposes a stochastic formulation of the pathfinding problem, assuming that obstacles of arbitrary shapes may appear and disappear at random moments of time. Moreover, we consider the case when the environment is only partially observable for an agent. We study and evaluate two orthogonal approaches to tackle the problem of reaching the goal under such conditions: planning and learning. Within planning, an agent constantly re-plans and updates the path based on the history of the observations using a search-based planner. Within learning, an agent asynchronously learns to optimize a policy function using recurrent neural networks (we propose an original efficient, scalable approach). We carry on an extensive empirical evaluation of both approaches that show that the learning-based approach scales better to the increasing number of the unpredictably appearing/disappearing obstacles. At the same time, the planning-based one is preferable when the environment is close-to-the-deterministic (*i.e.*, external disturbances are rare). Code available at https://github.com/Tviskaron/pathfinding-in-stochastic-envs.

# INTRODUCTION

Consider a mobile robot operating in a complex, non-stationary environment, *e.g.*, a service robot that has to transfer documents between the offices in an office building. One of the key challenges that arise when such robot navigates from its current location to the target one is that at no moment of time the robot possesses an accurate model of the environment. Among the main reasons for that the following can be named.

First, the *apriori* map of the environment (*e.g.*, the floor plan) is either unknown or approximate, *i.e.,* it does not contain the crucial information about the furniture, open/closed doors, etc. This leads to a necessity to invoke the so-called simultaneous localization and mapping (SLAM) (*Bresson et al., 2017*) pipeline that builds and constantly updates the map, based on the data from the sensors installed on the robot (cameras, lidars, *etc.*). Due to the measurement errors and to the limited sensors' range, the robot at each

Corresponding author
Alexey Skrynnik, skrynnik@airi.net

timestep acquires only a local semi-accurate patch of the map. Combining these patches to a global map *via* SLAM typically results in a more inaccurate map, which is subject to constant changes while the robot is progressing toward its goal.

Indeed, planning a path on the basis of such a map (that constantly changes and does not contain accurate information about a large portion of the environment) is a challenging task. Within a planning framework, it is typically addressed *via* re-planning, *i.e.,* at each timestep, an agent constructs a new plan taking into account the up-to-date map. This can be done in a straightforward fashion (from scratch) using, for example, a suitable variant of A* algorithm (*Hart, Nilsson & Raphael, 1968*), or *via* the more involved techniques, like incremental search, that re-use the search efforts of the previous planning attempts (*e.g.*, D*Lite (*Koenig & Likhachev, 2002*) algorithm, which is widely used in mobile robotics). Overall, such approaches can cope reasonably well with the problems induced by the robot's internal disturbances like noisy sensor data and partial observability.

There exists, however, a much harder problem, associated with the disturbances that are external to the robot. The environment (and, thus, its representation as a map) can change due to the actions of other agents interacting with the environment, *e.g.*, people that close/open doors, move pieces of furniture from one place to the other *etc.* These stochastic changes may lead to complete failures of the described planning/re-planning approach.

As an example, consider the environment, depicted at the top of Fig. 1. Even if it is fully observable, the robot can stuck in an oscillating behavior due to the stochastic (from the robot's perspective) blockage of one of the grid cells that can correspond to opening/closing a door. When the door is closed the robot constructs a detouring path and starts moving along it. After three steps, the door opens and a shorter path is found which is adopted by the robot (Fig. 1, top right). However, when the robot comes close to the door, it might be closed again, and the robot switches back to the detour path. Another challenging example is depicted on Fig. 1 at the bottom row. Due to partial observability and, again, unpredictable changes in the environment, the robot is stuck being unable to find a path to the goal on a map produced by accurately combining all observations.

We are unaware of the works that study the pathfinding problem when the environment is both partially observable and stochastic (when the obstacles may appear/disappear unpredictably). Our work aims to fill this gap. We develop and study empirically two orthogonal approaches to tackle the problem: the one that follows the planning/re-planning scheme (*Ghallab, Nau & Traverso, 2016*), and the learnable one, when we utilize a reinforcement learning approach (*Sutton & Barto, 2018*; *Moerland, Broekens & Jonker, 2020*) to optimize a policy function that maps observations to actions. Within the planning approach, we rely on a heuristic search based method that constantly attempts to find a path on the grid. As expected, this approach often fails due to the reasons explained above. To mitigate this issue to a certain extent we develop an extension of this approach that handles the history of observations in a way to heuristically identify stochastic obstacles and avoid them. Moreover, we apply a learning approach to the problem at hand and optimize a policy (deep neural network) that learns to deal with the stochasticity end-to-end. One can think of this as learning to adaptively change the action selection heuristics online

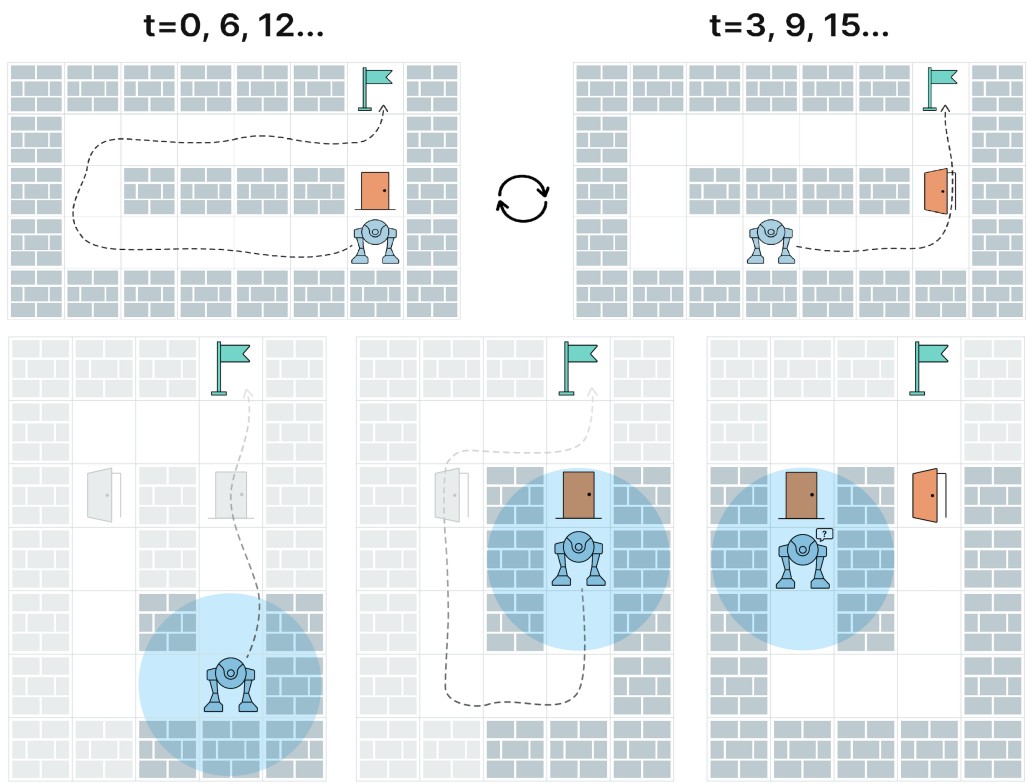

**Figure 1 The challenges associated with pathfinding in stochastic environments.** Top row: the robot with unlimited field of view is stuck in an oscillating behavior due to blocking/unblocking of a single grid cell (which might correspond to opening/closing the door by humans). Bottom row: due to the partial observability the robot is unaware that the previously blocked cell becomes traversable (the door opens) and, thus, the robot is not able to plan a path to the goal. Both cases can be successfully solved by the learnable policy suggested in the paper, which is empirically shown to learn such behaviors as "wait for an obstacle to disappear," "keep exploring the environment for additional options of reaching the goal", *etc.*

(when solving a pathfinding query). For example, the agent may choose to "wait" as many timesteps as needed when a passage gets temporarily blocked.

We carry out an extensive experimental evaluation on a wide range of setups and show that the designed learnable policy outperforms the planning approach both in terms of success rate and the solution cost (*i.e.,* number of actions needed to reach the goal) when the number of stochastically appearing/disappearing obstacles is not zero. Thus, we infer that the learning-based approaches should be the tool of choice when solving the studied kind of problems. Despite the latter may sound obvious, we are unaware of the works that empirically confirm this claim.

Overall, our main contributions can be summarized as follows. First, we introduce and study a challenging variant of the single-agent pathfinding problem inspired by the real-world robotic applications. In this setting the obstacles might stochastically appear/disappear in the environment and the latter is only partially-observable to the agent. Second, we propose the planning based and the learning based approaches to solve the problem. The latter utilizes reinforcement learning and is able (as shown

empirically) to form an adaptive pathfinding policy that successfully handles a wide range environment's disturbances. Finally, we empirically show that the designed learnable approach outperforms the planning one in the majority of the setups (on different maps, with different numbers of stochastic obstacles *etc.*).

The rest of the paper is organized as follows. Section 2 provides a brief overview of related works. Section 3 focuses on the formulation of the problem of pathfinding in stochastic environments with partial observability. Section 4 describes the methods of planning and asynchronous reinforcement learning that are being compared. Section 5 is devoted to the experimental study of the limits of applicability of the methods under consideration. In the conclusion, the results obtained are discussed.

## RELATED WORK

**Context**: The previous work in grid-based pathfinding was mainly focused on the application of the planning-based approaches to solving the problem. It was known that the planners based on the heuristic search are the versatile tools when the environment is static and partially observable. These planners have not been examined (both theoretical and empirical) in the setting with both unpredictably appearing/disappearing obstacles and partial observability, thus it was not evident whether planning approaches will succeed in it. On the other hand, pure reinforcement learning (RL) techniques (without hierarchy and model) were known to be very powerful in solving a wide range of problems with simple casual structure (like playing the video game of pong), however, when it comes to the problems that require reasoning about the outcomes of the series of actions (like navigation on a grid) RL shows much worse results. *e.g.*, in one of the most citepd papers, that addresses the navigation on a fully-observable grid without stochastic obstacles (*Panov, Yakovlev & Suvorov, 2018*), the RL methods demonstrated pure convergence of the learning process in even simplest cases. In some papers considering reinforcement learning in a partially observable stochastic environment, tabular methods are used that do not presuppose scaling to large-sized environments (*Pena & Banuti, 2021*). In the other recent paper from OpenAI (*Cobbe et al., 2020*), which describes a state-of-the-art RL benchmark including the grid-based navigation problem with the maximum grid size being $25 \times 25$, the advanced RL methods were not able to generalize easily and fast, which means that navigation queries on large grids remained unsolved. Overall, no clear evidence that RL methods, in general, can handle navigation problems on the large partially-observable grids exist so far.

**Relevant papers**: There exist a lot of different works relevant to the considered problem. Most of them are related to robotics, as the problem of operating in unknown environments with partial observability, uncertainty, and presence of the dynamic obstacles naturally appears in this field of research. *Fiorini & Shiller (1998)* introduces velocity obstacles—one of the major approaches to avoid dynamic obstacles that is based on the idea of predicting their further movement and choosing such an action that does not lead to a collision with any of the observable dynamic obstacles. This idea was further used in many approaches, including such algorithms as ORCA (*Van Den Berg et al., 2011*) and ALAN (*Godoy et al., 2018*), developed for multi-agent pathfinding problems.

Neural networks and machine learning are also widely used for planning in dynamic environments. In *Chen et al. (2020)* and *Zhu et al. (2014)*, biologically-inspired neural networks were applied for planning in unknown dynamic environments. Recent works, such as *Lei, Zhang & Dong (2018)* and *Wang et al. (2020)*, have tried to apply reinforcement learning to solve the problem of planning in dynamic environments. Moreover, it is also worth mentioning such kind of planners as ABT (*Kurniawati & Yadav, 2016*) or DESPOT (*Ye et al., 2017*) that solve the POMDP-problem, building a belief-state tree to deal with uncertainties and presence of dynamic obstacles.

Another field of research related to this work is heuristic search algorithms. Though having a lot of restrictions and assumptions to be applicable, they provide strong theoretical guarantees such as completeness and even optimality. *Koenig & Likhachev (2002)* intrudes state-of-the-art algorithm called D* Lite for planning in unknown partially-observable environments. In *Van Den Berg, Ferguson & Kuffner (2006)*, there was presented an anytime version of D* algorithm that works not only in unknown environments, but also can deal with dynamic obstacles. Another well-known approach is SIPP (*Phillips & Likhachev, 2011*) that can be applied for planning in the environments with dynamic obstacles and guarantees to find optimal solutions. However, it assumes that the environment is fully observable and trajectories of dynamic obstacles are known.

The stochastic shortest path (SSP) is a generalized version of the classical shortest path problem with a presence of stochasticity. In most cases, stochastic behavior is expressed in a non-deterministic result of the actions' execution. Mainly, it is considered as a Markov Decision Process (MDP) and the solution of such kind of problems is a policy that chooses which action to produce in any state to minimize the solution cost. An overview of different variations of this problem is given in *Randour, Raskin & Sankur (2015)*. Though the SSP problem contains stochasticity, it is assumed that the probability distributions are known, which makes it different from the problem that is considered in this paper.

It is also worth noting a direction of research where planning algorithms are combined with reinforcement learning. In *Skrynnik et al. (2021)* and *Davydov et al. (2021)*, the authors train RL agents in a centralized (QMIX) and decentralized (PPO) way for solving multi-agent pathfinding tasks. The resulting RL policies are combined with a planning approach (MCTS), which leverages the resulting performance. In *Ferber, Helmert & Hoffmann (2020)*, RL was used to learn the heuristic function to make it more informative. A similar idea of using reinforcement learning to get a better heuristic was suggested in *Micheli & Valentini (2021)* but for the problem of temporal planning.

Overall, there exist a large body of works that study the problems which are similar to ours in some aspects. However they all are different in the set of assumptions. As noted above, some of the works assume that the environment is known and fully observable, some of them—that even trajectories of dynamic obstacles are known. Most assume that the trajectories of the dynamic obstacles can be predicted at least for a short period of time. Approaches that deal with stochastic environments assume that probability distributions are given, so they can use them to build an optimal policy. Contrary to all these assumptions, in the problem that we are considering, the environment is changing in an unpredictable way.

## PROBLEM STATEMENT

Consider an agent moving on a 4-connected $M \times N$ grid. At each time step of the discrete timeline $T = 0, 1, \ldots, T_{max}$, where $T_{max}$ is the duration (length) of the *episode*, a grid cell can be either occupied or free. The cells that are occupied for all time steps are called static obstacles, the cells that are occupied for some time steps while being free for the others correspond to the stochastic obstacles (*e.g.*, closing doors, chairs that are moved by humans etc.). Indeed, the agent can use only the free cells for movement.

The action set for the agent is comprised of five actions: $A = \{up, down, left, right, wait\}$. Being at the grid cell $c$ at timestep $t$, an agent can opt to either *wait* at the current cell or to move to one of the cardinally-adjacent cells. Let $c'$ denote the target cell of the action, which is either the same cell or one of the neighboring ones. In case $c'$ is free at $t + 1$, the action is considered valid and the agent is transferred to $c'$. If, however, the destination cell of the move is blocked in the next time step the agent stays put, *i.e.,* is kept in its current cell. Each action $a \in A$ is associated with a non-negative cost: $cost(a) = w \in R_{>0}$. We assume this cost to be uniform, *i.e.,* 1, for all actions in the rest of the paper. In case the action chosen by the agent turns to be invalid the agent stays put but it still incurs a +1 cost.

The grid topology, its size and status of all the grid cells at a certain/any time step is not known to the agent. Instead, the agent can observe the grid environment only locally. Different ways to model this local observability model can be suggested. In this work we adopt the most easy-to-implement one: when located at the grid cell $c$ at time step $t$, the agent observes a $(2 \cdot R + 1) \times (2 \cdot R + 1)$ patch of the grid environment, which is centered at $c$, where $R$ is the given visibility range. In the example depicted in Fig. 2, $R = 5$, which means that the agent observes a $11 \times 11$ patch of the grid at each time step.[1] Within the visibility range the agent is able to observe the blockage status of the cells, however it is not able to distinguish which cells are blocked only temporarily (due to the appearance of the stochastic obstacles) and which are blocked constantly (due to the static obstacles).

We assume that the agent can not predict the blockage status of the grid cells which are both within or out of the visibility range. However, it is able to memorize the past observations if necessary.

The problem now is formulated as follows. Given the start and the goal location (cell) design a mapping from the (history of the) observations to actions, *i.e.,* the *policy* $\pi$, s.t. the chance of reaching the goal cell within the $T_{max}$ time steps is maximized. In this work we are not restricting ourselves to design the policy that minimizes the cost of reaching the goal, *i.e.,* the sum of costs of the actions that led to the goal cell, however obtaining the lower cost paths is, obviously, preferable.

[1] The introduced observation model allows the agent to "see through the obstacles". Although this assumption is not realistic in the majority of the real-world cases it, indeed, reflects the property of the local observability and at the same time is very easy to implement and experiment with.

## METHODS

To solve the considered problem we investigate two approaches. The first one relies on finding a path on the grid and then applying the first action of this path. The second approach is based on reinforcement learning. Here the agent optimizes a policy that maps the observations to actions end-to-end and follows this policy in a reactive fashion. Next, we describe both approaches in more details.

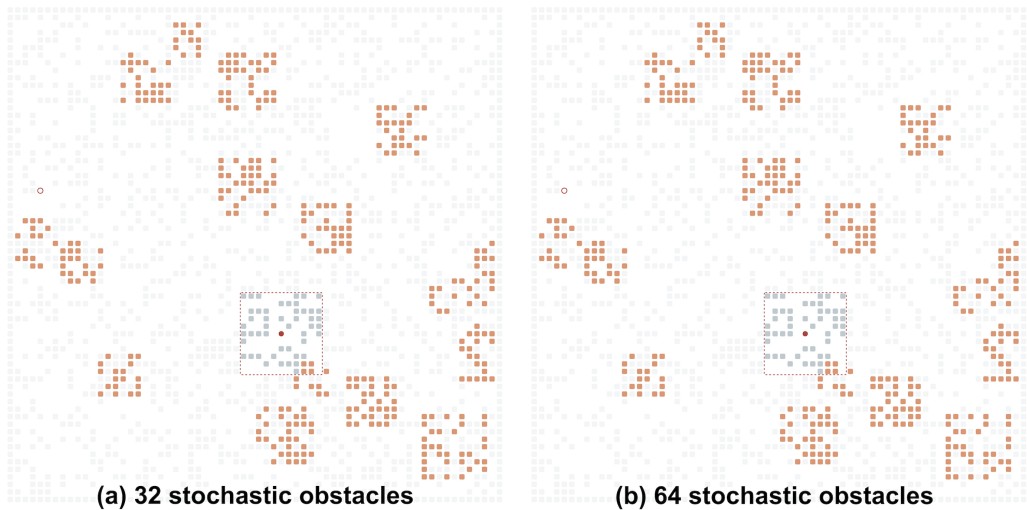

**(a) 32 stochastic obstacles**     **(b) 64 stochastic obstacles**

**Figure 2** **Examples of the grid environments with different numbers of the stochastic obstacles (shown in orange).** (A) Thirty-two stochastic obstacles; (B) 64 stochastic obstacles. Undiscovered static obstacles are transparent. The agent's field of view is shown by the red square.

## Planning

The main idea of planning is to repeatedly (i) construct a full sequence of actions that reach the goal state (*i.e.,* the grid cell) from the current one (which is the start cell initially), (ii) apply the first action of the plan.

When constructing a plan, we rely on the history of the received observations, *i.e.,* we memorize the observations and construct a map out of the map. At each time step, upon receiving the new observation we update the map. We use it then to find a path from the current cell to the goal one by applying a pathfinding algorithm. We then extract the first move action from this path and add it to the resultant plan. If no path is found, we opt to perform a greedy action that moves the agent closer to the goal (more details on this will be given below). The high-level pseudocode of the planning algorithm is presented in Algorithm 1.

The presented algorithm relies on the following intrinsic assumptions. First, it is assumed that the agent knows the goal's coordinates within a reference frame, initially centered at the start position of the agent, and is able to localize itself within this frame at each timestep. Second, the agent is able to combine all the observations into a single representation of the environment, *i.e.,* the map. This map is built/updated incrementally. The size of the map is unknown.

The crucial procedure of the algorithm is *PathFinding*, which finds the path on the map provided with the start and goal locations, where start constantly changes due to agent moving through the environment. The most prominent way of solving pathfinding problems in the environments with partial observability is D*Lite algorithm (*Koenig & Likhachev, 2002*). Instead of re-planning the path from scratch after applying each action, it extensively reuses the previously built search tree to speed up the search. Unexpectedly,

**Input**: start cell $s$, goal cell $g$, initial observation $obs$, maximal number of actions the agent can make $T_{max}$

**Output**: either *success* or *failure*

$map \leftarrow obs$

$c \leftarrow s$

$t \leftarrow 0$

**while** $t < T_{max}$ **do**

> **if** $c = g$ **then**
>
> > | return *success*;
>
> **end**
>
> $obs \leftarrow GetCurrentObservation()$
>
> $map \leftarrow UpdateMap(map, obs)$
>
> $path \leftarrow PathFinding(c, g, map)$
>
> **if** $path = \varnothing$ **then**
>
> > | $a \leftarrow GreedyAction()$
>
> **else**
>
> > | $a \leftarrow FirstActionFromPath(path)$
>
> **end**
>
> $ApplyAction(a)$
>
> $t \leftarrow t + 1$

**end**

return *failure*

**Algorithm 1:** High-level algorithm of reaching the goal in a stochastic environment via planning.

---

**Input**: number of training epochs $E_{max}$

**Output**: policy $\theta$

$\mathcal{D} \leftarrow \varnothing;\ e \leftarrow 0;\ h_0 \leftarrow \varnothing;$

$\theta \leftarrow InitializeActor()$

$\phi \leftarrow InitializeCritic()$

**while** $e < E_{max}$ **do**

> $\mathcal{D} \leftarrow GenerateTrajectories();\ \theta \leftarrow UpdateActor(\mathcal{D}, \phi)$
>
> $\phi \leftarrow UpdateCritic(\mathcal{D})$

**end**

return $\theta$

**Algorithm 2:** A policy optimization algorithm (training phase).

our preliminary tests have shown, that the performance of D*Lite is worse compared to the sequential re-planning with A* from scratch after each move. The main reason for such phenomenon is that at some (numerous) time steps the feasible path from the agent's current position to the goal is not existent due to the stochastic obstacles that temporarily block the narrow passages. In case these blockages appear close to the agent, running A* from scratch detects unsolvability notably faster compared to D*Lite which actually

**Input**: initial observation *obs*, maximal number of actions the agent can make $K_{max}$
**Output**: trajectory $\mathcal{D}$
$\mathcal{D} \leftarrow \varnothing;\ k \leftarrow 0;\ h_0 \leftarrow \varnothing$
**while** $k < K_{max}$ **do**
    **if** *EpisodeIsDone*() **then**
        return $\mathcal{D}$
    **end**
    $action, h_k \leftarrow GetAction(obs, h_{k-1})$
    $ApplyAction(a)$
    $obs \leftarrow GetObservation()$
    $r \leftarrow GetReward()$
    $\mathcal{D}.AddTuple(o, a, r)$
    $k \leftarrow k + 1$
**end**
return $\mathcal{D}$

**Algorithm 3:** An algorithm for generating actions with a learnable policy model (inference phase).

plans backwards from the goal state. As such blocking happens often (especially when the number of stochastic obstacles is high) sequential invocation of A⋆ actually proved to be beneficial in our preliminary tests and, thus, we adopted A⋆ to implement *PathFinding* in this work.

Additionally, we attempted to adapt the *UpdateMap* procedure to the environments with stochastic obstacles. Recall, that the agent is not able to distinguish which grid cells within its visibility range are blocked temporarily, due to the stochastic obstacles, and which cells are blocked for good by the static obstacles. The basic *UpdateMap* procedure treats all the blocked cells as the static obstacles, adding them to the map. This ignores the fact that some of the temporarily blocked cells might become free in future. To this end, we introduce a modified *UpdateMap* procedure that tries to detect stochastic obstacles leveraging the history of observations. More specifically, in case when the agent observes a cell that is currently blocked but was free according to the preceding observations, it is marked as blocked temporarily. When this cell is within the observation radius its actual blockage status is taken into account while finding a path. When the agent moves away and this cell is no longer within the visibility range it is considered to be free, so the path can go through it. Intuitively, this allows the agent to anticipate that the the previously seen stochastic obstacle might move away. This variant of the planning algorithm with the additional indication of the stochastic obstacles is called Stochastic A⋆ (SA⋆).

Both regular planning algorithm (A⋆) and the adapted one (SA⋆) share the following additional techniques: exploration threshold and greedy action. Both these techniques are tailored to handle the case when the path to the goal does not exist (due to the temporal presence of the stochastic obstacles). Recall, that we assume that the size of the map is not known to the agent. This infers that it is not technically possible to detect that the current pathfinding query is unsolvable as the search algorithm will keep on exploring the

environment assuming that the portion of the environment, that has not been seen/mapped before, is traversable. To mitigate this issue we impose the threshold on the number of internal iterations of A*/SA*. When this threshold is exceeded the search is aborted with the *no-path-found* result. In such case instead of picking the random action we choose the one that is likely to move the agent closer to the goal—the so-called greedy action. It is chosen as follows. When A*/SA* terminates due to the exploration threshold without finding the path we pick the node in the search tree that corresponds to the cell which is the closest to the goal. We then reconstruct the path to this cell in the tree and pick the first action of the path.

## Learning

The main idea of the reinforcement learning (RL) approach is to optimize a policy $\pi$, which maps the observation to an action. The policy is trained to maximize the cumulative expected reward (cost function) for each interaction episode. We use the partially observable setting since the agent has no access to the global state.

The advantage of RL over planning approaches is that the agent can learn the adaptive heuristic of acting in an environment with stochastic dynamics. At the same time, it should be taken into account that the work of the learnable approaches is divided into two phases: the actual training on prepared environment configurations (training phase, see Algorithm 2) and the work of the trained model on any environments (inference phase, see Algorithm 3).

Formally, in RL setting, the interaction of an agent with the environment is described as partially observable Markov Decision Process (POMDP), which can be described as tuple $(S, O, A, P, r, \gamma)$, where $S$ is the set of environment states, $o \in O$ is a partial observation of the state, $a \in A$ is the set of agent's actions, $r(s, a) : S \times A \to \mathbb{R}$ is a reward (cost) function, and $\gamma$ is the discount factor. The agent has no access to states $S$ (true coordinates and full obstacle map), and the policy is a mapping from observations to actions: $\pi(a|o) : A \times O \to [0, 1]$.

We propose and describe an end-to-end architecture to train the agent in grid pathfinding scenarios. And we believe that our learning approach is applicable for a wide range of pathfinding tasks. The scheme of the learning approach is presented in Fig. 3.

As already mentioned in Section 3 in the proposed environment the observation space $O$ of the agent is a multidimensional matrix: $O : 2 \times (2 \times R + 1) \times (2 \times R + 1)$, that represents the part of the environment around the agent within radius $R$. It includes the following two matrices.

- *Obstacle matrix*: 1 encodes an obstacle, and 0 encodes its absence. If any cell of the agent's field of view, which is outside the grid, is encoded as an obstacle. The agent does not distinguish between the type of obstacles. Both static and stochastic obstacles are encoded the same.
- *Target matrix*: if the agent's goal is inside the observation field, then there is 1 in the cell, where it is located, and 0 in other cells. If the target does not fall into the view, then it is projected onto the nearest cell of the observation field (*Skrynnik et al., 2021*).

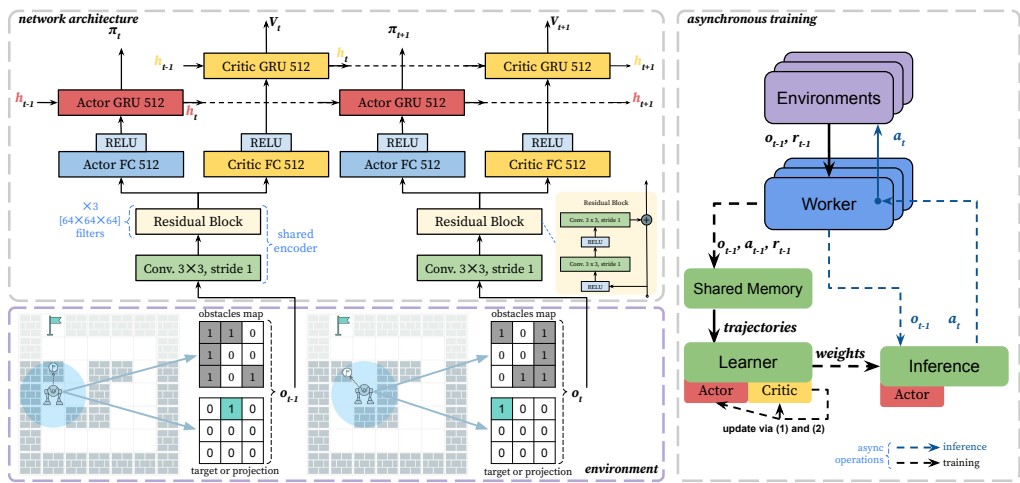

**Figure 3** **The scheme of the proposed asynchronous learning approach.** The bottom-left part shows the environment and observation encoding. The top-left part of the scheme shows the neural network architecture. We use residual layers as a shared encoder and recurrent heads. The recurrent layers (GRU) are responsible for remembering obstacles and environment dynamics. The right part shows the asynchronous learning procedure. The green blocks show GPU computations.

At any time step, the agent has five actions available: stay in place, move vertically (up or down), or move horizontally (right or left). The agent can move to any adjacent free cell.

The agent receives a reward of 1.0 when it reaches the goal and 0.0 in all other cases. We have chosen this function so one can train the agent, which can deal with a wide range of tasks in partially observable grid environments with stochasticity.

To learn a policy in a model-free setting, there are a number of well-known methods in reinforcement learning, which can be roughly divided into two classes—value-based (*Mnih et al., 2015*) and policy-based (*Schulman et al., 2015*; *Haarnoja et al., 2018*; *Lillicrap et al., 2016*) methods. The first group of approaches is characterized by the use of replay buffer to store experience and can learn only deterministic policies. The second group of methods is specifically designed for operating stochastic policies. In the problem we are considering, which is characterized by stochastic behavior of the environment itself, it is necessary to provide an opportunity to work with probability distributions on a set of agent actions. On-policy methods (*Schulman et al., 2017*) that use only current experience are the most promising for the partially observed formulation of the pathfinding problem. This is due to the fact that in some cases of recurrent stochasticity in the environment, it is necessary to use a stochastic policy, which is most effectively learned by on-policy policy gradient methods . Also, this makes it possible to improve the quality of state prediction by observation when using recurrent neural network models.

We optimize the policy $\pi_\theta$, which is approximated by a neural network $\theta$, using Proximal Policy Optimization (PPO) method (*Schulman et al., 2017*). The approach is a variant of the actor-critic algorithm, and proven effective in many challenging domains (*Berner et al., 2019*; *Yu et al., 2021*; *Cobbe et al., 2020*). To adapt PPO for the POMDP setting, we approximate the state $s_t$ using a recurrent neural network (RNN): $s_t \approx f(h_t, o_t)$, where

$h_t$ is a hidden state of RNN. PPO uses clipping in objective to improve performance monotonically. The clipped objective penalize the new policy $\pi_{\theta_{k+1}}$ for getting far from the previous one $\pi_{\theta_k}$.

To approximate policy and value we adapt network architecture from IMPALA (*Espeholt et al., 2018*). As a feature encoder we use residual layers. We have changed the original architecture and removed max pooling, similar to AlphaZero architecture (*Silver et al., 2017*), which used akin encoding for observations. Removing max pooling is crucial, to prevent loosing spatial information of the grid observation. After the shared feature encoder, there are recurrent layers separate for the actor and the critic.

We use single GPU asynchronous training setup based on SampleFactory (*Petrenko et al., 2020*). We called the modified version of the PPO algorithm as asynchronous proximal policy optimization (APPO). The asynchronous training can be divided into two main parts, which run in parallel: accumulating new trajectories and policy updating. The policy used to collect experience may lag behind the current one. That discrepancy is called a policy lag, and it negatively affects the performance. The policy lag especially affects the learning of recurrent architectures. To stabilize training for such case IMPALA (*Espeholt et al., 2018*) introduced importance sampling for value targets, called V-trace, which we also use in our training.

## EXPERIMENTAL EVALUATION

### Environment

We designed and implemented the environment simulator that takes all the specifics of the partially observable pathfinding and stochastic obstacles into account. The following parameters are related to the interaction between the agent and the environment: observation radius (agents observe $1 \leq R \leq Size$ cells in each direction) and the maximum number of steps in the environment before the episode ends $Horizon \geq 1$.

The stochastic part of the environment is described with the following parameters: a number of the stochastic obstacles $\geq 0$; obstacle size $[x, y]$ (we assume that each stochastic obstacle is a square whose size is in between $[x, y]$); obstacle density $\in (0, 1]$ (the chance of each cell comprising the stochastic obstacle to be blocked, *i.e.,* some of the cells forming the obstacle can be free); obstacle move radius $\geq 0$ (defines how far from the original placement the obstacle can move); obstacle appear time range $[x, y]$ (the range of steps during which a stochastic obstacle can be active); obstacle disappear time range $[x, y]$ (the range of steps during which a stochastic obstacle can be inactive). Figure 2 shows examples of the environments with the defined stochastic obstacles.

For the experimental evaluation, we have set the following environment parameters of stochastic obstacles: obstacle size = [5,10], obstacle density = 0.7, obstacle move radius = 5, obstacle appear time range = [8,16], obstacle disappear time range = [8,16]. All the values for the parameters with ranges are chosen randomly out of the corresponding range. In other words each stochastic obstacle is a square with a side of five to ten, and 70% of the cells forming this square are blocked. The structure of each stochastic obstacle does not change with time. However, every tick of time it can move , but not more than five

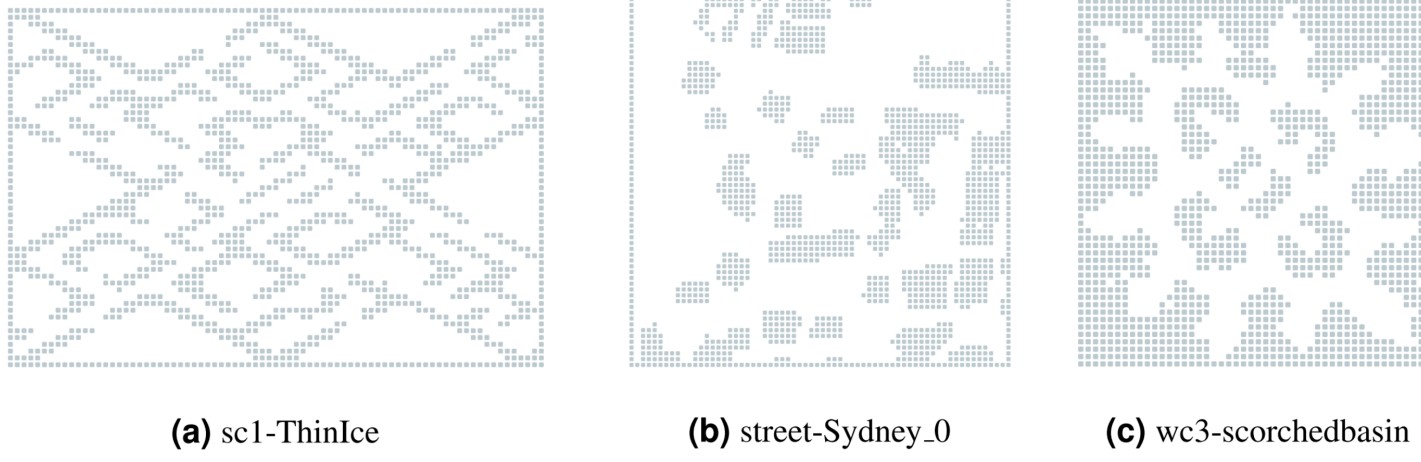

**(a)** sc1-ThinIce      **(b)** street-Sydney_0      **(c)** wc3-scorchedbasin

**Figure 4** **Examples of the evaluated maps from different collections.** (A) sc1-ThinIce; (B) street-Sydney_0 ; (C) wc3-scorchedbasin.

cells from the initial position. A stochastic obstacle is present on the map from 8 to 16 time ticks, after which it disappears for a period of 8 to 16 time ticks and then reappears.

It is also worth to note that stochastic obstacles can be superimposed on static obstacles, on top of each other, but cannot block the cell in which the agent is located.

Moreover, the observation radius was set to five and *Horizon* to 512.

## Setup

During the experiments, we evaluated both planning approaches –the basic one, *i.e.,* A⋆, and the improved one, *i.e.,* SA⋆. There is only one crucial parameter that is needed to be set for them –a maximum number of iterations, that was set to 10000. Besides the planning approaches, we have also evaluated the learning one, *i.e.,* APPO. The details about its training process and the choice of hyperparameters are described in Section 5.3.

The experiments were conducted on the maps from three different collections taken from MovingAI (*Sturtevant, 2012*)—a grid-based pathfinding benchmark: (a) *wc3*—36 maps from Warcraft three computer game; (b) *sc1*—75 maps from Starcraft computer game; (c) *streets*—30 maps with real city data taken from OpenStreetMap. The chosen maps represent different landscapes with varying topological structure, *i.e.,* they include maps with small passages, large open areas, prolonged obstacles of non-trivial shapes etc. The original maps can be in size up to $1024 \times 1024$. For our experiments we have scaled them to $64 \times 64$ or such a size that the lowest side is 64. These collections were divided for training and test subsets in a ratio of $8 : 2$, simply using the alphabetic names of the maps. Examples of the maps are shown in Fig. 4. The names of all the maps that were used for tests can be found in Section 5.4.

For each of the evaluated maps, there were generated 200 instances. The instances were generated randomly, but in such a way that the path between start and goal locations is guaranteed to exist for the static obstacles. Moreover, each of these instances was evaluated with a different number of stochastic obstacles: from 0 to 200 with an increment of 25.

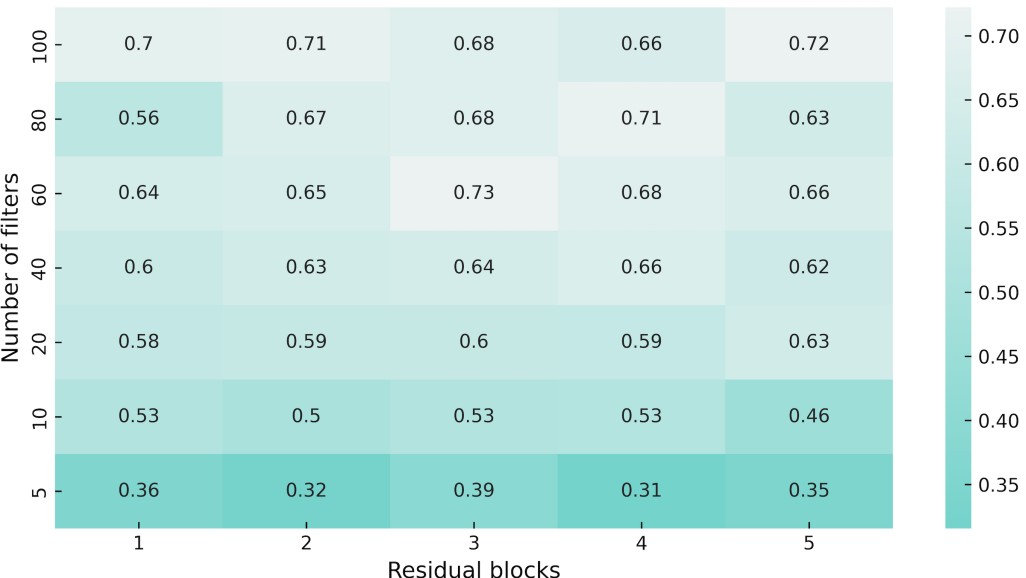

**Figure 5** **Heatmap of the network architecture parameters.** We used a version with 60 filters and three residual blocks as encoder in APPO model for all experiments, since it showed better performance in our hyperparameter sweep.

During the experiments, we evaluated such a parameter as success rate, which shows the ratio between the number of successfully solved instances to their total amount. An instance is considered successfully solved in case the solution was found within less than 512 steps (*Horizon* parameter), *i.e.,* the resulting plan contains less than 512 actions. Besides the success rate, we have also evaluated the average episode length, taking *Horizon* value for the instances that were not solved by an algorithm, and computed how many times each of the algorithms has found a solution with less number of actions.

To evaluate the computational efficiency of the approaches, we have measured such a parameter as "steps per second" (SPS), which shows how fast the approach returns an action that the agent needs to perform on the current step. The more actions the approach returns within a second, the faster it works. The evaluation of all algorithms were conducted using AMD Ryzen Threadripper 3970X CPU (single-core). Also, the performance of RL approach, *i.e.,* APPO, were tested using NVIDIA GeForce RTX 3080 Ti.

## APPO training

First, we made a hyperparameter search to adjust the number of residual blocks and the number of filters in them (see Fig. 5). As the environment configuration, we use a map with 30% density of static obstacles and 64 stochastic obstacles in $64 \times 64$ grid, obstacle size is $\{5, 10\}$, the density of the stochastic obstacles is 0.7, appear and disappear time ranges are $\{8, 16\}$. The agent was trained for 100 million steps. The best results were shown by a network with three residual layers and 60 filters in them, thus we use these settings in all proceedings experiments.

Second, we trained APPO for one billion steps using training collection. For each episode, the initial position of the agent and his target, as well as the configuration of stochastic obstacles, were sampled randomly. Solving a task with a large number of stochastic obstacles is difficult in terms of exploration. Thus, we used curriculum learning to automatically adjust the difficulty of the environment. In each curriculum phase, the agent is trained until it reaches the average success rate of 0.9 on the 256 consequent episodes, after that the number of stochastic obstacles is increased by one. The final number of stochastic obstacles at the end of the training was 46.

## Results

The aggregated results of the experiments are shown in Table 1. As one can see in most of the cases, APPO solves more instances than planning approaches. The only case where APPO shows a worse success rate is the one that doesn't contain any stochastic obstacles, where planning approaches have solved 100% of instances. The success rate of A* algorithm, which doesn't detect stochastic obstacles and remembers all the obstacles it has seen, is very poor. While APPO and SA* successfully solved 45% of the instances with 200 stochastic obstacles, A* has shown even worse success rate with only 25 stochastic obstacles. Such behavior explains by the fact that A* in most cases can't pass stochastic obstacles and fails to find a path to the goal. This behavior also explains its high SPS—A* makes a very few expansions before it makes a conclusion that the path cannot be found.

A more detailed view of the success rates of the approaches is presented in Fig. 6. It shows the success rates of the approaches on each of the maps separately. As one can see, there is not a single map where SA* significantly outperforms APPO. Despite the points with 0 stochastic obstacles, in all the cases they either show very close results, or APPO outperforms SA*. There are some maps, for example, `timbermawhold` or `swampofsorrows`, that show significantly higher success rates compared to other maps. Such behavior explains by the fact that these maps contain several relatively small disjoint areas. As a result, the instances on these maps are much simpler than on the other maps, as the distance between start and goal location is much lower. On the other hand, there are some maps, where success rates of all approaches drop to less than 20% on the instances with the highest amount of stochastic obstacles. There are actually two reasons for such behavior. First, there are maps such as `ThinIce` (see Fig. 4A) or `Typhoon`, that contains a difficult structure for partially observable environments, *i.e.,* they contain "trap" areas, that are on the way to the goal but do not actually lead to it. Second, some maps of `wc3` collection in the original MovingAI benchmark contain huge borders of obstacles, that affected the scaled maps and reduced their actual size with traversable areas to $48 \times 48$ (see Fig. 4C). Thus, the density of stochastic obstacles on such maps is actually much higher than on other maps.

To get some insight about the quality of the found solutions, we have computed how many times APPO or SA* has found a solution with less number of actions. We have excluded the results of A* in this comparison, as its success rate is very low. However, there actually were some instances where A* has found a solution with the least number of actions—220 out of 56,000. The results were aggregated among the collections and are presented in Fig. 7. As one can see, there actually presents two more lines called "Equal"

**Table 1  Averaged results of A\*, SA\*, and APPO aggregated over all the evaluated maps.** Bold values highlight better performance (episode length–lower better, success rate–higher better) for each number of stochastic obstacles. SPS denotes steps per second.

| Algorithm | Obstacles | SPS (CPU) | SPS (GPU) | Episode length | Success rate |
|---|---|---|---|---|---|
| A\* | | 1127.87 | – | **57** | **1** |
| APPO | 0 | 105.58 | 812.3 | 95.64 | 0.93 |
| SA\* | | 1052.73 | – | **57** | **1** |
| A\* | | 2226.9 | – | 331.14 | 0.38 |
| APPO | 25 | 100.71 | 754.58 | **120.11** | **0.93** |
| SA\* | | 847.81 | – | 135.39 | 0.88 |
| A\* | | 2589.9 | – | 410.22 | 0.21 |
| APPO | 50 | 100.39 | 731.58 | **157.29** | **0.9** |
| SA\* | | 866.21 | – | 189.96 | 0.8 |
| A\* | | 2438.91 | – | 438.91 | 0.15 |
| APPO | 75 | 98.23 | 712.26 | **200.41** | **0.82** |
| SA\* | | 909.81 | – | 232.68 | 0.72 |
| A\* | | 2226.41 | – | 453.42 | 0.12 |
| APPO | 100 | 100.04 | 702.45 | **242.72** | **0.74** |
| SA\* | | 955.12 | – | 271.56 | 0.65 |
| A\* | | 2061.56 | – | 462.88 | 0.1 |
| APPO | 125 | 98.49 | 693.69 | **279.28** | **0.65** |
| SA\* | | 976.84 | – | 297.88 | 0.59 |
| A\* | | 1911.56 | – | 468.22 | 0.09 |
| APPO | 150 | 98.23 | 686.08 | **309.25** | **0.57** |
| SA\* | | 988.06 | – | 320.56 | 0.54 |
| A\* | | 1793.01 | – | 474.08 | 0.08 |
| APPO | 175 | 97.75 | 667.67 | **334.54** | **0.51** |
| SA\* | | 1021.84 | – | 338.94 | 0.49 |
| A\* | | 1631.31 | – | 475.79 | 0.07 |
| APPO | 200 | 98.39 | 659.66 | **353.37** | **0.45** |
| SA\* | | 1014.4 | – | 354.45 | **0.45** |

and "Failed". "Equal" line indicates the portion of instances that were successfully solved by both approaches with an equal number of actions, while "Failed" one indicates the portion of instances that were solved neither by APPO nor by SA\*. The results on sc1 and street collections are very similar. The portion of instances that were solved with equal number of steps on sc1 and street collections is relatively small, while on wc3 it's much higher. This behavior on wc3 is explained by the presence of such maps as timbermawhold and swampofsorrows with isolated parts, where the instances are easy and thus solved by both approaches with equal cost. About 70% of all the instances on these maps were successfully solved by both approaches with an equal number of actions. The behavior of SA\* and APPO methods shows that the planning approach outperforms the others when the number of stochastic obstacles is very small. The learning approach outperforms the others when the number of stochastic obstacles rises to about 100. When the number of stochastic obstacles rises to the maximum, both approaches show close results. Generally,

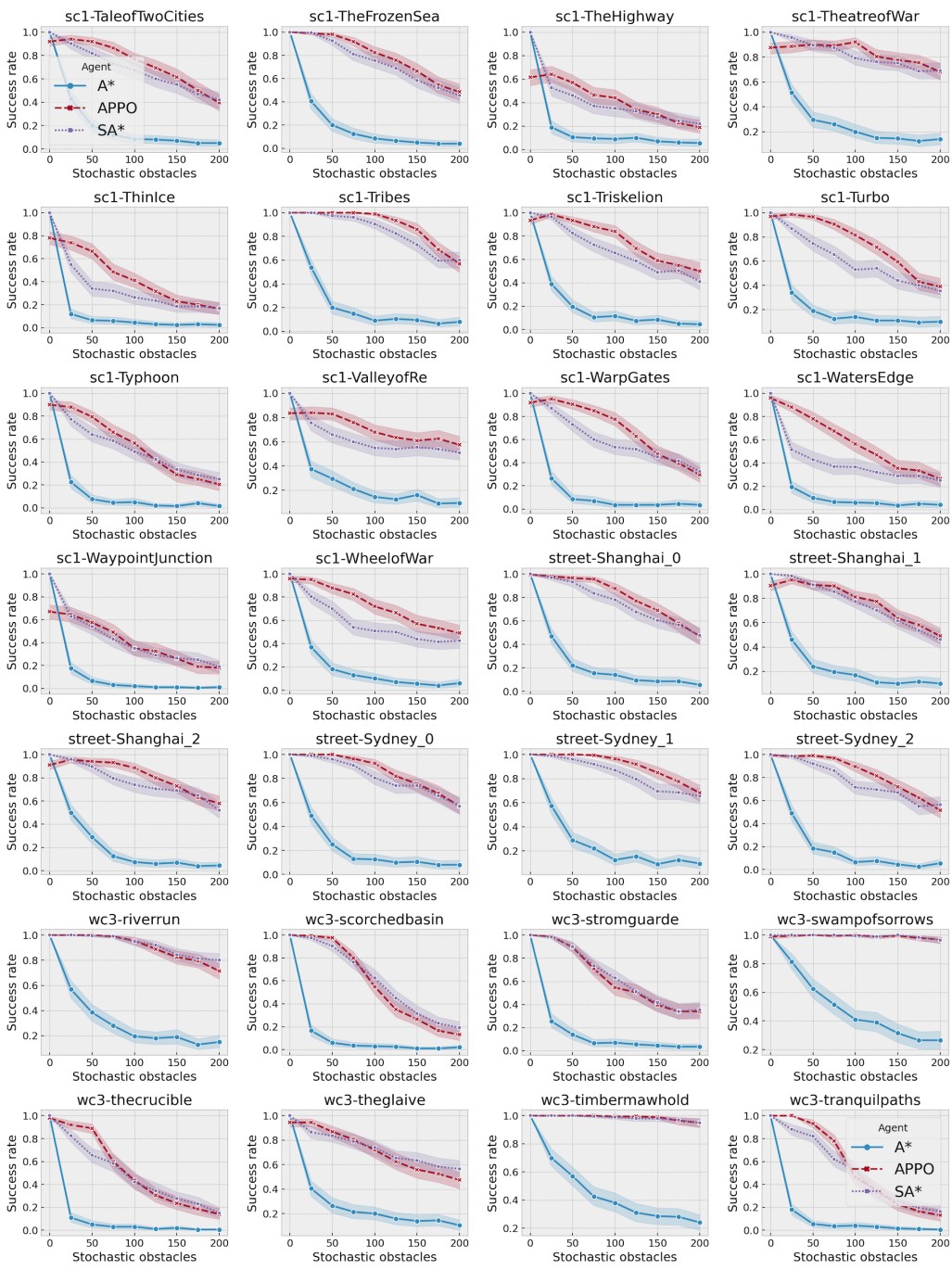

**Figure 6** **Success rates of A\*, SA\* and APPO on each of the evaluated maps depending on the number of stochastic obstacles.** The shaded area shows 95% confidence interval.

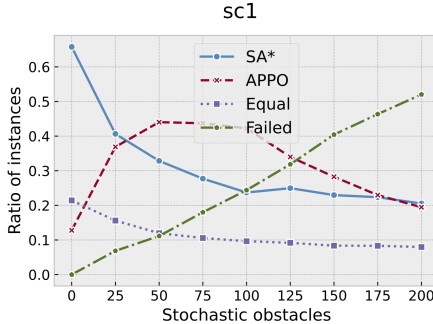
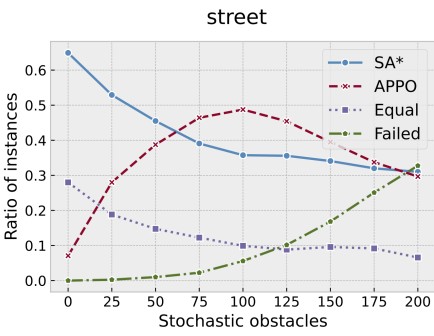
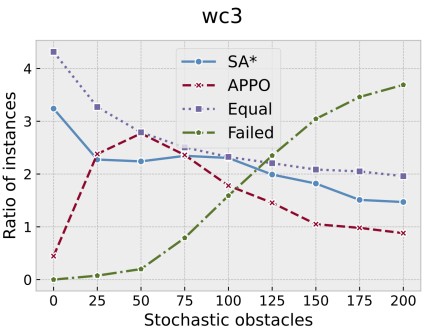

**Figure 7** **Comparison of the solutions found by SA⋆ and APPO depending on the collection of the maps and number of stochastic obstacles.** Points on "SA⋆" and "APPO" lines indicate the portions of instances for which the corresponding approach has found a solution of a better quality than other one. "Equal" correspond to the portion of instances for which both approaches have found solution with equal quality. "Failed" correspond to the portion of instances that were solved neither by SA⋆ nor by APPO.

these plots indicate the same trends as the ones that show the success rates of the algorithms.

## CONCLUSION AND DISCUSSION

In this article, we have introduced and studied a challenging variant of the single-agent pathfinding problem inspired by real-world robotic applications. In this setting, some of the obstacles unpredictably appear/disappear in the environment, and the latter is only partially observable to the agent. We designed two orthogonal approaches to solve this problem: planning-based and learning-based. For the former, we utilized the well-known A⋆ algorithm and suggested its modification, called Stochastic A⋆ (SA⋆), that differs in the way how the incoming observations are processed; for the latter, we have proposed an original asynchronous policy optimization method (APPO) based on the established actor-critic neural network architecture. Both approaches were experimentally evaluated on a range of setups involving different maps and degrees of stochasticity (*i.e.,* numbers of the appearing/disappearing obstacles). The results indicate that both of the suggested approaches has their own pros and contras. SA⋆ is evidently faster than APPO but its success rate is generally lower compared to APPO. The only case when SA⋆ performs better/on par with APPO is either when no stochastic obstacles are present at all or when this number is very high. Both cases can be seen as the outliers. For all other configuration APPO, indeed, is able to successfully solve more instances than SA⋆. We believe that this happens due to the ability of APPO to adaptively adjust the heuristic of choosing actions, which is learned rather than hard-coded. We also would like to note that low computational efficiency of our implementation of APPO is not a fundamental problem, as there is a room for a significant speed-up *via* using specialized code implementations (*e.g.*, TensorRT).

One of the perspective avenues for future research is investigating the analogous problem statements, but when certain predictions about the dynamics of the environment can be made. One of such settings that is of particular interest is the decentralized multi-agent pathfinding setting, when each agent can distinguish between the static obstacles and the moving agents and is able to predict, to a certain extent, the future moves of the

latter. We assume that in such settings, the planning-based approaches may exhibit a better performance due to the additional knowledge that they can take into account, *i.e.,* the locations that will be blocked at the next time step due to the moves of the other agents. In such case, pathfinding algorithms can be straightforwardly extended to reason about the temporal dimension and to build plans that will avoid future collisions with the other agents. As for the learning-based approaches, modifying(and learning) them for such settings might be more problematic. Indeed, such approaches for decentralized multi-agent pathfiding do exist currently, see (*Sartoretti et al., 2019*; *Riviere et al., 2020*) and others, but mainly they rely on the accurate long-horizon predictions of how the other agents will behave, *i.e.,* they rely on the ability to acquire/accurately reconstruct the full paths of the other agents to their goals. In case this ability is limited, *e.g.,* only the next action can be inaccurately predicted, their performance might get worse. Therefore, the question on whether the learning-based approaches will beat the planning-based ones in such settings is still to be answered.

### Funding
This work was supported by the Ministry of Science and Higher Education of the Russian Federation under Project 075-15-2020-799. The funders had no role in study design, data collection and analysis, decision to publish, or preparation of the manuscript.

### Grant Disclosures
The following grant information was disclosed by the authors:
The Ministry of Science and Higher Education of the Russian Federation under Project 075-15-2020-799.

### Competing Interests
The authors declare there are no competing interests.

### Author Contributions
- Alexey Skrynnik conceived and designed the experiments, performed the experiments, analyzed the data, performed the computation work, prepared figures and/or tables, authored or reviewed drafts of the article, and approved the final draft.
- Anton Andreychuk conceived and designed the experiments, performed the experiments, analyzed the data, performed the computation work, prepared figures and/or tables, authored or reviewed drafts of the article, and approved the final draft.
- Konstantin Yakovlev conceived and designed the experiments, analyzed the data, authored or reviewed drafts of the article, and approved the final draft.
- Aleksandr Panov conceived and designed the experiments, analyzed the data, authored or reviewed drafts of the article, and approved the final draft.

### Data Availability
  Python code is available in the Supplemental Files.

## Supplemental Information

Supplemental information for this article can be found online at http://dx.doi.org/10.7717/peerj-cs.1056#supplemental-information.

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
