# Peer review of "Pathfinding in stochastic environments: learning vs planning"

_PeerJ Computer Science, doi:10.7717/peerj-cs.1056_

## Round 0.1 · original submission · Major Revisions

Some concerns are raised by the reviewers. Please address them and provide a detailed response letter. Thanks.

Reviewer 1 ·

Basic reporting

Language quality is good but can be improved in some parts. Please consider using a spellchecker and let your collegues read it:

- Abstract, 3rd Line: "..., assuming that the obstacles of..." -> "..., assuming that obstacles of..."
- Page 1, Line 30: "(i.e. the map), external disturbances..." -> "(i.e. the map) and external disturbances..."
- P3, L94: "In Conclusion..." -> "In the conclusion,..."
- Please make it consistent: You write "Fig. 1", but "section 1". Decide for either writing style.
- P6: The indention of Algorithm 3 is farther to the right than the other ones
- P7, L 204: "unexpectedly" -> "Unexpectedly"
- P7, L213: "we have tried". This is very wordy and should be rephrased
- P7, L216: "Vanilla" -> "The basic"
- P7, L216: "...same way just adding" -> Wordy and colloquial, please rephrase
- P7, L229: "that the we" -> "that we"
- P7: Algorithm 3 is referenced before Algorithm 2
- P7, L249: "Formally, in RL settings" -> "Formally, in RL, setting"
- P7, L255: "describe end-to-end" -> "describe an end-to-end"
- P8, L280: "In our opinion" -> What is the reasoning?
- P9, L301: Espeholt was already cited in L291
- P9, L306: "The following parameters...", please rephrase by adding an "and" instead of ";"
- P9, L320: "- basic one" -> "- the basic one"
- P9, L330: "An examples" -> "An example"
- P10, L365: "Success rate" -> "The success rate"
- P10, L368: "just" -> please rephrase. Wordy and colloquial
- P10, L372: "there is no a map" -> "there is not a map"
- P11, L390: "As one can see..." -> Please rephrase this sentence, as I don't understand it
- P11, L399: "shows its best" -> "outperformes the others"
- P11, L400: "Learning approach" -> "The learning approach"
- P11, L400: "shows its best" -> "outperformes the others"
- P11, L405: "the obstacles might stochastically" -> "the obstacles stochastically"
- P12, Table1: Please state in the caption what the bold entries mean
- P14, L418: "is somewhat" -> wordy, please rephrase

- P1, L29-31. I think the latter is contained in the former. Please clarify.
- Consider writing references in brackets, e.g., (Doe et al., 2022). It can improve the readability.

- P3, L98. You state that there are no studies on stochastic obstacles. Do you mean combined with unknown environments, or in general? For the latter, there is, e.g.:
Zhang, Y., Jun, Y., Wei, G., & Wu, L. (2010). Find multi-objective paths in stochastic networks via chaotic immune PSO. Expert Systems with Applications, 37(3), 1911–1919. https://doi.org/10.1016/j.eswa.2009.07.025

- P8, L271: Please elaborate on the multi-agent version

Experimental design

- P4, Problem Statement: Can the robot see through walls or other obstacles? How does it relate to an actual real-world robot?
- Problem Statement: Please make it more formal to enable others to reproduce your results.
- P7, L210: Is there a boundary (in terms of number of stochastic obstacles) when D*Lite gets worse?
- P9: Please formalise the environment generation more. Assign proper variables to the properties.
- P9: How does "number of obstacles" relate to "obstacle density"?
- P9, L322: Why 10000?
- P9, L329: How did you scale these grid-based maps? I think there are some challenges.
- P10, L343: Please find a different name for "frames per second", this could be misleading

Validity of the findings

- P10, L345: Please justify why you chose a GPU and CPU approach, as the comparison seems unfair.
- P10, L373: Please elaborate on your statistical testing, e.g. what kind of test did you use?
- P11, L401-402: You state, that the performance of both approaches is close when the number of stochastic obstacles rises to maximum. Please elaborate more on that, as this is an important finding.

- P14, L415: You state that the learning approach outperformes the planning one with increasing stochasticity. However, on P11, L401 you say, that the results are close with a high number of obstacles. How does that fit together?

- P14: Please extend you future outlook.

Reviewer 2 ·

Basic reporting

The literature references could be discussed better in terms of what they mean. E.g., ll 119, 120, ... references are simply listed, and l 147 simply declares 'they have crucial differences'. This needs to be elaborated - in what sense are they different?

Googling for the keywords, it seems that the article "Reinforcement learning for pathfinding with restricted observation space in variable complexity environments" AIAA2021-1755 by Pena & Banuti addresses reinforcement learning for pathfinding with limited observation space in dynamic/stochastic environment, in which obstacles may move without discernible pattern for the agent. Clearly, the present article is much more thorough, on the other hand, the expressed statement that no one has ever looked into this seems inaccurate.

Experimental design

There appears to be a discrepancy between the problem statement l 175 that the agent is able to memorize past observations, and l 281 in which only the current experience is considered the most promising.

l 382 mentions 'traps' in the maps, but declares they are only an issue with stochastic obstacles, but it seems that dead ends that are larger than the observation space would always form traps for the learned approach?

The evaluated maps are limited to hand-designed maps that were intended to be navigated, particularly in video games. It is not clear to me if that implies a bias and thus a limitation of the study. This should be discussed.

The study equates 'more stochastic' with 'higher number of stochastic obstacles'. I think this is too simplistic. From the definition in the article, an obstacle is stochastic if it changes its state a single time. Is an environment in which 200 obstacles appear in step 1 and remain afterwards then more stochastic than an environment in which 190 obstacles turn on and off randomly throughout the run? Further, from the perspective of the agent, this means that every obstacle is potentially stochastic, as we don't know whether it will change its state later.

Validity of the findings

l244-l246 states that learning approaches are more successful because they can learn the stochastic behavior. This seems like a truism. Further, does this contradict l281, that the agent does not rely on previous experiences concerning the stochastic obstacles?

I am not sure I agree with l415, regarding Fig. 7: the way I understand it is that in all three cases, for # of stochastic obstacles >= 200, SA* outperforms APPO. If that is not the case, please revise the explanation of what this means. (also, all captions in Fig. 7 have the same typo 'obstatcles')

Additional comments

Overall a very nice and well written paper!

---

## Round 0.2 · accepted · Accept

The authors have addressed the comments in the revised version. It can be accepted.